# Position: Unlabeled $\neq$ No Human Supervision in Visual Learning

**Dong Lao** [1]

## Abstract

This position paper argues that the absence of labels does not imply the absence of human supervision in visual learning, and urges the research community to identify sources of supervision more explicitly. Many recent methods in computer vision build upon representations learned from large-scale unlabeled data, and are therefore grouped under the same umbrella term "unsupervised." However, different data curation schemes and training objectives embed substantially different human priors on which models rely, and we argue that one "unsupervised" umbrella term is no longer capturing these distinctions. This ambiguity makes it harder to compare unsupervised learning research conducted under different assumptions, coinciding with a sharp decline in papers titled with "unsupervised" in flagship computer vision conferences since 2021, despite continued growth of the field. While we fully embrace pre-training as a strong foundation for modern computer vision, we advocate for a community-level effort toward greater conceptual clarity: authors are encouraged to disclose priors in data selection and learning objectives, and to specify which components of a learning pipeline depend on which assumptions. Standardized disclosure practices can improve academic communication, ensure fairer comparisons, and preserve methodological diversity in unsupervised learning.

## 1. Introduction

Computer vision research has recently welcomed a transformation driven by the success of large-scale visual representation learning. Models trained on massive collections of images and videos, often regarded as foundation models (Chen et al., 2020a; He et al., 2020; Radford et al., 2021;

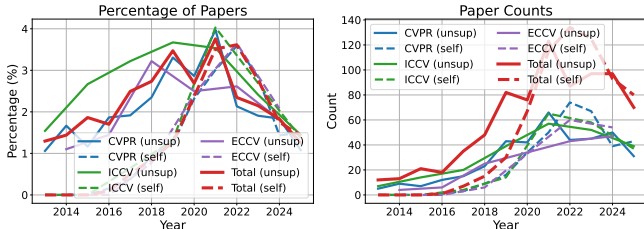

*Figure 1.* **How "unsupervised" learning is framed over time.** Title-based statistics from flagship computer vision conferences. Solid lines denote paper titles containing *unsupervised*, while dashed lines denote *self-supervised*. Both terms peak around 2021 and decline thereafter despite continued growth in total publications. This surprising trend draws our attention to a broader shift in how unsupervised learning is positioned within the field.

Caron et al., 2021; He et al., 2022), now yield highly transferable features that serve as default backbones across a wide range of downstream tasks. This pre-training and transfer learning paradigm (Kornblith et al., 2019) has substantially reduced task-specific supervision and has reshaped how model pipelines are designed. Methods[1] that build upon such representations are frequently described as unsupervised[2] due to the absence of human-provided labels.

While this framing is accurate, it obscures an important conceptual distinction: the absence of human labels does not imply the absence of human supervision. Decisions surrounding data curation and preprocessing introduce substantial human priors that learning objectives can critically depend on. For example, ImageNet (Deng et al., 2009) organizes semantic classes around WordNet synsets (Miller, 1995), while modern vision-language datasets (Schuhmann et al., 2022) depend strongly on metadata design and curation choices (Xu et al., 2024). Beyond data, such priors are also embedded in learning paradigms. For example, contrastive learning encodes assumptions about what con-

[1]Division of Computer Science and Engineering, Louisiana State University, Louisiana, USA. Correspondence to: Dong Lao <dong.lao@lsu.edu>.

*Proceedings of the 43rd International Conference on Machine Learning*, Seoul, South Korea. PMLR 306, 2026. Copyright 2026 by the author(s).

---

[1]We deliberately refrain from citing specific research that makes unsupervised claims in this manuscript, and focus on common patterns and practices across the broader research community.

[2]Self-supervised learning is typically regarded as a subset of unsupervised learning, as it needs no human annotations. It differs from classical unsupervised approaches, e.g. K-means (Lloyd, 1982), by explicitly constructing supervision signals from the data. In the context of this paper, we adopt unsupervised learning as an umbrella term encompassing self-supervised methods, while acknowledging the conceptual distinctions between these paradigms.

stitutes a meaningful visual entity (i.e., an "object"), and thus which variations should be treated as invariant (Wang & Isola, 2020; Tian et al., 2020; Xiao et al., 2021). In their training data, when an object is centered in an image, it reflects a photographer's intention to frame it into a "shot" and a dataset creator's intention to include such an image. Even though no labels are used directly, these human intentions still supervise models. The absence of labels, therefore, does not imply the absence of human supervision.

This issue is not inherent to the definition of unsupervised learning itself, but rather emerges from a historical shift in how unsupervised learning is practiced and deployed. Prior to the rise of large-scale pre-training, unsupervised learning was predominantly *in-domain* (Olshausen & Field, 1996; Hinton & Salakhutdinov, 2006; Kingma & Welling, 2013; Bengio et al., 2013). Models learned from digits were applied to digits (LeCun, 1998), models learned from faces were applied to faces (Karras et al., 2019). In such settings, the relevant data-distribution assumptions were explicit and localized. However, modern pre-training operates at a vastly different scale and is routinely adopted across domains. This shift introduces a new and under-examined challenge: priors embedded in the pre-training pipeline are no longer confined to a single data distribution, but are implicitly transferred across settings. Assumptions that were once explicit and localized become generic and often invisible. This conceptual shift has manifested at the community level. We analyze flagship computer vision conference publications in the past decade, and notice two patterns: (1) unsupervised (and self-supervised) learning becomes less explicitly foregrounded after its peak around 2021-2023; (2) papers described as "unsupervised" increasingly rely on pre-trained representations (Fig. 1 and 2). This trend indicates a community-level shift in how unsupervised learning is positioned and practiced. In the remainder of this paper, we present hypotheses that may explain this shift and discuss their implications for the future of unsupervised learning.

Importantly, we do not challenge the widely accepted definition of unsupervised learning, nor do we diminish the role or effectiveness of pre-training. Instead, we seek to draw attention to an emerging consequence of this success. As unsupervised learning shifts from in-domain to foundation-scale, the role of inductive priors embedded in pre-training becomes increasingly difficult to identify, track, and reason about. At the same time, under the shared umbrella term of "unsupervised" learning, the dominance of foundation models makes it difficult for new unsupervised paradigms to compete under comparable computational scale, thereby discouraging early-stage exploration and threatening methodological diversity. We therefore advocate for greater conceptual clarity in how unsupervised learning claims are made and evaluated. Specifically, we encourage authors to explicitly disclose data-selective biases underlying their methods,

and to specify which components of a learning pipeline are label-free and which depend on human supervision or certain data priors. We believe that clarifying what is, and is not, assumed by "unsupervised" methods is essential for sustaining exploratory research beyond dominant technical paradigms, and for attributing why working methods succeed to the appropriate components of the learning pipeline.

## 2. Empirical Signal from Flagship Venues

To ground the discussion, we analyze how unsupervised learning has been framed in flagship computer vision venues. We begin with a title-based signal: whether authors describe their work as "unsupervised" or "self-supervised" in the title. This signal is intentionally coarse because it reflects how authors choose to foreground unsupervised learning without requiring us to adjudicate individual methodological claims. To address the possibility that titles merely reflect naming conventions, we further complement this analysis with a full-text scan of *CVPR* papers and an analysis of pre-training dependence among papers titled with "unsupervised."

### 2.1. Observation

**Title-based analysis.** We collected main conference paper titles of *CVPR*, *ICCV*, and *ECCV* between 2013 and 2025. *CVPR* is held annually, while *ICCV* and *ECCV* alternate on a biennial schedule, such that each year includes two flagship conferences. For each venue and year, we count papers whose titles contain *unsupervised*, excluding cases of *unsupervised domain adaptation*, which represent a distinct research problem from unsupervised learning. This exclusion does not mean that domain adaptation is unrelated to our argument. Rather, it reflects a difference in problem formulation. In domain adaptation, distribution shift is itself an explicit object of study (Ganin et al., 2016; Yang et al., 2020; Wang et al., 2020; Park et al., 2024; Chung et al., 2025). In contrast, our focus is on claims where data-distribution assumptions are often inherited implicitly through pre-training. Further, to account for the possibility that observed trends reflect a shift in naming conventions, we perform the same analysis for titles containing "self-supervised" or "self-supervision." All counts are normalized by the total number of accepted papers.

Several trends are immediately apparent from Fig. 1. First, both "unsupervised" and "self-supervised" papers increase steadily throughout the late 2010s, reaching their highest proportions around 2021. This period coincides with the birth of large-scale self-supervised pre-training. Second, and surprisingly, the proportion of paper titles explicitly using either term declines after this peak, despite continued growth in total paper counts. For example, at *CVPR*, the fraction of papers labeled "unsupervised" peaks at nearly 4% in 2021 and drops to close to 1% by 2025.

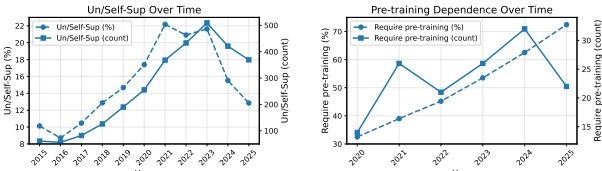

*Figure 2.* **Trends of un/self-supervised papers and pre-training dependency in *CVPR*.** Left: the full-text Un/Self-Sup trend is consistent with the title-based analysis, declining after its peak around 2021-2023. Right: among *CVPR* papers titled with "unsupervised," reliance on pre-trained features increases consistently since 2020.

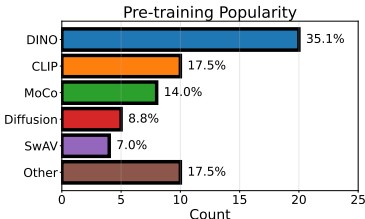

*Figure 3.* **A small set of pre-trained backbones dominates.** Aggregating all years, we show the distribution of explicitly named pre-training backbones used by pre-trained-dependent papers in our corpus. A few pre-training families account for a large share.

**Full-text analysis.** Given such observation, we extended this analysis to the full-text level over all 17,435 *CVPR* papers from 2015 to 2025. We scraped the corresponding manuscripts from the official CVPR website and used the Qwen-2.5-32B-Instruct (Qwen et al., 2025) language model to analyze whether each method explicitly relies on a named pre-trained model. This procedure does not infer hidden dependencies, but only records author-stated pre-training reliance. This fine-grained full-text signal follows the same qualitative pattern as the title-based analysis: it rises from 10.13% in 2015 to 22.17% in 2021, remains high through 2023, and then declines to 15.52% in 2024 and 12.85% in 2025 (Fig. 2). This finding further supports the trend that unsupervised and self-supervised learning are becoming less explicitly foregrounded in the *CVPR* literature.

**Pre-training dependency of "unsupervised" papers.** We further analyze all 287 *CVPR* papers from 2020 to 2025 whose titles explicitly contain "unsupervised." For each paper, we *manually* inspect whether the proposed method explicitly relies on a pre-trained model external to the paper's core contribution. This manual inspection also serves as the ground truth to evaluate the accuracy of automated dependency extraction used to support the analysis. Against the manual annotations, the automated classification achieves 91.5% precision, 81.8% recall, and 86.9% accuracy.

As shown in Fig. 2, among "unsupervised" papers, reliance on pre-training external to the paper's core contributions increases steadily over this period. Fig. 3 adds another complementary signal: this pre-training dependence is concentrated in a small number of backbone families. DINO (Caron et al., 2021; Oquab et al., 2023; Siméoni et al., 2025), CLIP (Radford et al., 2021), MoCo (He et al., 2020; Chen et al., 2020b; 2021), diffusion models (Rombach et al., 2022; Podell et al., 2022), and SwAV (Caron et al., 2020) account for a large share of the explicitly named pre-training sources (Bommasani, 2021). This concentration matters because when many "unsupervised" pipelines inherit the same few pre-trained sources, observed downstream properties may reflect shared upstream curation and inductive biases rather than novel emergent principles.

## 2.2. Hypotheses for the Observed Shift

These statistics are descriptive rather than causal. Here, we outline three complementary hypotheses that together may explain why unsupervised and self-supervised learning have become less explicitly foregrounded.

**Hypothesis 1: Comparison pressure.** Work under weaker assumptions can appear less competitive when evaluated under the same "unsupervised" umbrella term, especially when compared at the same scale. This pressure may be amplified when scaling up data volume, model size, or pre-training compute yields more immediate performance gains than developing new unsupervised or self-supervised methods from scratch. This mechanism is consistent with the trend in Fig. 2, where a growing share of papers with "unsupervised" in the title build on top of large-scale pre-trained models.

**Hypothesis 2: Framing of unsupervised contributions.** The way researchers frame their contributions may be shifting. As pre-training becomes an infrastructural component, unsupervised learning is increasingly treated as an implicit ingredient rather than an explicit contribution. Many papers now build on state-of-the-art pre-trained encoders and focus on downstream applications, which reduces the incentive to emphasize an unsupervised or self-supervised aspect in titles. In this sense, the title-based trends in Fig. 1 can be interpreted as unsupervised learning becoming an implicit component rather than an explicit focus.

**Hypothesis 3: User-side indifference.** A third perspective arises from end users of these models. For practitioners and consumers, whether a model is trained in a supervised or self-supervised manner is often secondary to empirical performance, availability, and ease of integration. As large pre-trained vision foundation models become de facto building blocks (Liu et al., 2021; Kirillov et al., 2023; Assran et al., 2023), their training paradigm becomes less salient to users. This user-side indifference can further reduce pressure to explicitly label methods as unsupervised.

Taken together, these hypotheses suggest two broader trends. First, pre-training has become widely adopted and deeply embedded in standard pipelines, while the assumptions and

priors that underlie these models are increasingly implicit. Second, fundamental unsupervised learning research that does not rely on large-scale pre-training may be becoming less accessible to researchers without substantial computational resources, and current publication incentives may provide limited positive feedback for such work. This tension motivates the need for clearer distinctions and more explicit communication about the forms of supervision and assumptions that methods rely upon.

# 3. Risks of Ambiguity

In this section, we outline several such risks and discuss how they can affect research practice and scientific interpretation. Before proceeding, we emphasize that the goal of this section is not to critique individual methods, nor to adjudicate the correctness of specific claims in prior work. We deliberately refrain from examining individual papers. Our focus is on the community-level risks that arise from ambiguity in how unsupervised learning is defined and communicated, independent of any particular implementation.

**Misattribution of Emergent Behavior.** A primary risk is the misattribution of emergent behavior in downstream pipelines. Many properties attributed to an "unsupervised" method, such as semantic discovery, objectness, generalization, or robustness to certain perturbations, may in fact be inherited from pre-trained representations rather than arising from the proposed mechanism itself (also see (Geirhos et al., 2020)). When pre-trained features embed strong priors, downstream methods can appear to exhibit emergent properties when the method is otherwise *exploiting* these inherited biases. In other words, priors from data curation and assumptions made by pre-training might be interpreted as emergent. This ambiguity can lead to overstated claims about the novelty of downstream learning mechanisms and hides the source of empirical findings.

**Erosion of Conceptual Clarity.** When inherited biases are not clearly separated from learned behavior, conceptual clarity suffers. Methods relying on fundamentally different assumptions are grouped under the same "unsupervised" label, despite operating in qualitatively different settings. Over time, the term "unsupervised" loses its ability to convey how a method learns, what assumptions it relies on, and where it is expected to generalize. This erosion of clarity complicates the interpretation of learning mechanisms and makes it harder to compare methods on equal footing.

**Distorted Research Incentives.** Ambiguity in attribution can also distort research incentives. When downstream performance gains are achieved primarily by leveraging increasingly powerful pre-trained representations, there is less incentive to study learning mechanisms that operate under weaker or more controlled assumptions. Methods may be favored even when their primary contribution lies in adapting to specific priors inherited from pre-training rather than introducing new learning principles. As a result, researchers may be incentivized to tailor methods to particular state-of-the-art pre-trained models and anticipate empirical gains. This dynamic can encourage incremental adaptation over fundamental innovation (Sculley et al., 2018).

**Reduced Methodological Diversity.** Over time, these incentives can contribute to reduced methodological diversity. Approaches that do not rely on large-scale pre-training, or that do not exploit certain data priors, may struggle to compete on benchmarks and consequently receive less attention. This risks marginalizing early-stage research on alternative supervision signals, e.g., physical interactions, motion, etc., or on relaxed data distribution assumptions, that do not benefit from the same priors. Such narrowing of the methodological landscape is particularly concerning for unsupervised learning, which historically serves as a frontier for exploring diverse supervision signals.

**Barriers to Same-Scale Evaluation.** Finally, ambiguity around supervision can raise barriers to fundamental exploration. Widely adopted pre-trained models (e.g., as in Fig. 3) often require substantial computational resources, and when unsupervised learning is implicitly equated with the ability to exploit such models, comparison at the same-scale settings becomes more difficult. This dynamic can limit who is able to contribute new ideas, especially within the "*GPU-poor*" academic settings, and slow progress toward research not tied to scale.

**Summary.** Taken together, these risks suggest that more explicit disclosure is necessary not only for accurate scientific exchange but for sustaining a healthy research ecosystem. By disentangling inherited priors from pre-training, the community can better evaluate progress, preserve methodological diversity, and advance unsupervised learning.

# 4. How Supervision Enters Without Labels

## 4.1. Data Distribution

A model trained without labels can nonetheless be shaped by human priors on the data distribution. Decisions made during data collection, curation, and preprocessing determine which visual patterns are prevalent, which are rare, and which are absent (Zeng et al., 2024). This point is closely related to the long-standing literature on dataset bias. (Torralba & Efros, 2011) showed that visual datasets are not neutral samples of the world: models trained on one dataset often fail to generalize cleanly to another dataset nominally designed for the same recognition task. More recent work revisits this issue in the era of larger datasets and modern networks, showing that dataset identity remains highly recoverable from images even after years of dataset

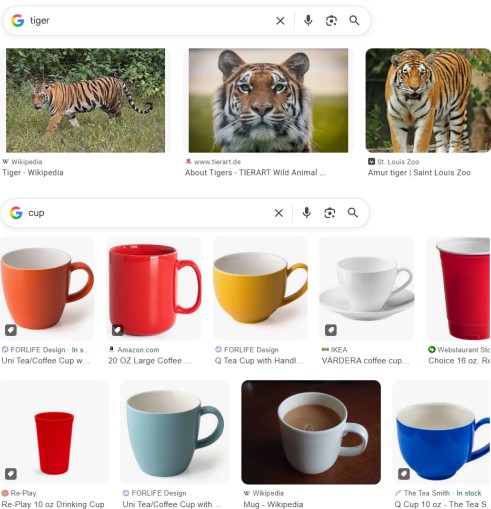

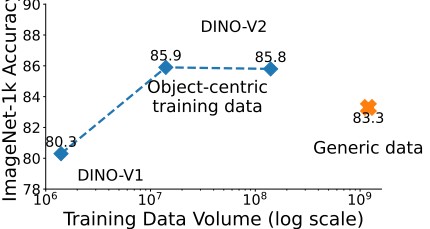

*Figure 5.* **Object-centric data dependency.** ImageNet-1k linear evaluation performance as a function of training data volume for DINO-family models (Caron et al., 2021; Oquab et al., 2023). Performance improves with scale under object-centric training data, but drops when training on more generic data.

*Figure 4.* **Human priors in web-scale visual data.** Top image search results for the queries "tiger" and "cup" returned by a standard Google Images search without manual filtering. Tiger images predominantly exhibit canonical poses, viewpoints, and centered framing, while images of cups frequently display handles oriented toward the right-hand side. These regularities reflect human preferences and selective biases rather than emergent properties of learning algorithms, illustrating how strong supervision can be embedded in data distributions even in the absence of explicit labels.

At the same time, not all useful priors in label-free learning should be reduced to human bias. Some priors arise from geometry, motion, physical interaction, cross-modal latent structure, or temporal structure (Achille et al., 2018; Schölkopf et al., 2021; Assran et al., 2023). For example, self-supervised monocular depth and depth-completion methods exploit geometric and sensor-consistency constraints (Yang et al., 2024; Wong & Soatto, 2019; Wong et al., 2020; Liu et al., 2022; Lao et al., 2024b) and transfer to downstream semantic tasks. Motion-based object discovery exploits independently moving entities that are separable (Xie et al., 2022; Lao et al., 2025). In embodied settings, manipulation concepts can be discovered from unlabeled demonstrations or multimodal streams using informativeness, mutual information, temporal abstraction, or closed-loop policy learning rather than human semantic names (Liu et al., 2024; Zhou & Yang, 2024; Zhou et al., 2025; Liu et al., 2025). These examples do not weaken our argument; they clarify it. The relevant question is not whether a prior is human, geometric, causal, temporal, or embodied, but whether the source of that prior is made explicit when a method is described as unsupervised.

construction effort (Liu & He, 2025). Related benchmarks further show that recognition systems remain sensitive to re-collected test sets, object pose and viewpoint, corruptions, and contextual co-occurrence patterns (Recht et al., 2019; Barbu et al., 2019; Hendrycks & Dietterich, 2019; Singh et al., 2020). These observations support our view: an unlabeled dataset is not neutral, and its biases, such as semantic categories, viewpoints, and lighting conditions, are often shaped by humans. Importantly, these choices encode human judgments about relevance given the specific intention and context during dataset construction (Dulhanty & Wong, 2019; Luccioni & Rolnick, 2023; Gavrikov & Keuper, 2024). Decisions about what constitutes a valid image, which scenes are worth capturing, and which variations are acceptable reflect assumptions about what the dataset is intended to represent. Even in the absence of labels, such judgments embed supervision into the data distribution itself, shaping learning indirectly but powerfully.

Even in the absence of deliberate or heavy-handed data filtering, web-scale data collection (Schuhmann et al., 2022; Byeon et al., 2022) alone introduces strong human priors. As illustrated in Fig. 4, common web queries already produce highly regularized visual patterns, such as canonical object poses, centered compositions, and consistent orientations, even when no manual curation is applied. These regularities arise upstream of any learning algorithm and are inherited by models trained on such data.

### 4.2. Learning Objectives

Beyond the data distribution itself, many modern self-supervised learning objectives implicitly rely on human priors for their validity. Contrastive methods provide a canonical example. They assume that different augmented views of the same sample correspond to the same underlying entity, while different samples correspond to distinct entities. Under this assumption, enforcing invariance across augmentations becomes a meaningful learning signal that encourages semantic representations.

This assumption holds reliably in curated, object-centric datasets (Deng et al., 2009), where images are designed to contain a dominant entity and where common augmentations preserve that entity's identity. However, it does not generalize universally (Xiao et al., 2021). In scenes with multiple interacting entities, stochastic textures, or

fine-grained scientific imagery such as microscopy or medical scans, the notion of a dominant object may be ill-defined. More broadly, work on invariance, disentanglement, and object-centric learning shows that such assumptions are often necessary for meaningful structure to emerge from unlabeled data (Achille & Soatto, 2018; Locatello et al., 2019; 2020a;b). Fig. 5 illustrates how self-supervised learning objectives depend critically on data distribution. When the training data scale is increased by incorporating more generic, non-object-centric images, the performance of DINO models degrades. This drop indicates a mismatch between data priors and the learning objective: the latter continues to enforce invariances that are no longer valid.

When learning objectives depend on assumptions that hold only under specific data regimes, supervision is not eliminated but displaced into the interaction between the dataset and the objective. Label-free learning, therefore, also remains shaped by human choices in learning objective design. For instance, horizontal flipping is a reasonable invariance for generic object recognition, but becomes harmful for road sign recognition, where flipping can invert semantic meaning. Such choices reflect human judgments about which variations should be treated as invariant. The same concern also applies beyond contrastive objectives. Different self-supervised objectives introduce different assumptions into the learning process, and these assumptions can be inherited by downstream components even when no human labels are used. Reconstruction-based methods provide a useful example: masked autoencoders achieve strong ImageNet performance (He et al., 2022), but reconstruction objectives can emphasize features that are not necessarily the most informative for perception (Balestriero & Lecun, 2024). These differences are visible in practice but difficult to quantify: different pre-training objectives, despite having similar pre-training performance, can produce substantially different feature distributions (see visualizations in (Lao et al., 2024a)) and downstream behavior. Recognizing these dependencies is essential for interpreting when and why unsupervised learning succeeds or fails.

## 5. Our Proposal: Clarifying and Broadening Unsupervised Learning Claims

Rather than redefining unsupervised learning or prescribing a particular technical direction, we propose a set of lightweight, constructive practices aimed at improving conceptual clarity and evaluation fairness. The proposal follows directly from the central claim of this paper: if supervision is no longer localized only in labels, but is distributed across data curation, pre-training, augmentation design, and evaluation choices, then these sources of supervision should be made explicit. Our goal is not to constrain methods or discourage the use of strong priors, but to ensure that claims

of unsupervised learning remain informative about the assumptions and dependencies under which a method operates. By making these assumptions explicit, we seek to preserve the empirical strengths of modern visual learning while enabling more meaningful comparison and interpretation.

Table 1 operationalizes this proposal as a minimal checklist for making supervision-relevant dependencies explicit. The checklist is deliberately lightweight: it is not intended as a complete taxonomy of all possible priors, nor as a rigid compliance mechanism. Instead, it identifies the main intervention points through which human supervision can enter otherwise label-free pipelines. The proposed items are orthogonal to contribution type: they do not penalize the use of pre-training, curated data, or strong invariances, but help distinguish *label-free* learning under different, often subtle but consequential, assumptions.

Although the checklist is textual, it is compatible with more quantitative diagnostics. Prior work has shown that tasks and datasets can themselves be embedded, compared, or differentiated through learned representations (Achille et al., 2019; Dukler et al., 2021). Such tools suggest a possible longer-term direction: disclosure could eventually be complemented by diagnostics that measure how much a method depends on particular data distributions, tasks, or upstream representations.

**Distinguishing Pre-training-Dependent Methods.** We first propose explicitly distinguishing methods whose core functionality depends on pre-training from those that learn directly from domain-specific data. We argue that methods whose success fundamentally depends on pre-trained representations should be described in a way that reflects this dependency, rather than being grouped under a single "unsupervised" umbrella. This distinction is not intended to diminish their contributions, but to clarify what aspects of learning are being addressed. Making this distinction explicit enables fairer comparison across approaches.

**Explicit Disclosure of Required Priors.** We further propose that papers claiming unsupervised learning explicitly disclose the priors required to support their learning paradigm. Authors should be encouraged to articulate which assumptions are essential and which are incidental, and to explore the extent to which these assumptions can be relaxed without fundamentally breaking the method. Importantly, such disclosure should include a discussion of data selective biases, regardless of whether labels are used.

**Robustness Across Pre-training Regimes.** Relatedly, we propose that unsupervised methods relying on pre-trained features should test across pre-training regimes. For example, a method may be evaluated across contrastive, reconstruction-based, predictive, or multimodal pre-trained encoders. Demonstrating consistency across such regimes

| Disclosure dimension | What to explicitly state |
|---|---|
| Pre-training dependence | Whether any pre-training is used (Y/N); backbone(s); approximate compute scale. |
| Frozen vs. adaptive reliance | Whether pre-trained components are frozen, partially adapted, or fully fine-tuned; which layers. |
| Pre-training data scope | Pre-training dataset size and modality; public vs. private data; domain overlap with downstream task. |
| Data distribution priors | Data source(s); filtering or curation rules; object-centric, viewpoint, or scene-selection biases. |
| Learning invariances | Augmentations applied; invariances enforced; known failure modes or violated assumptions. |
| Evaluation of leakage risks | Whether hyperparameters or representations in pre-training are tuned on/for evaluation data. |

*Table 1.* **Proposed disclosure checklist for label-free visual learning claims.** The goal is not to restrict methods, but to make supervision-relevant assumptions explicit and comparable. This disclosure is orthogonal to contribution type: it does not penalize the use of priors or pre-training, but clarifies where supervision enters pipelines described as label-free.

provides evidence for a generalized learning principle. Conversely, if a method performs well only when paired with a specific pre-trained backbone, this dependency should be acknowledged and discussed. Such clarification prevents overstatement and improves scientific transparency.

**Valuing Assumption Relaxation.** We also argue that unsupervised learning research should place greater value on relaxing assumptions, not solely on achieving peak benchmark performance. Under the current umbrella of unsupervised learning, methods operating under weaker assumptions are often compared directly to those exploiting stronger priors. However, relaxing assumptions is itself a meaningful scientific contribution. Methods that reduce object-centric bias, weaken augmentation invariances, or avoid reliance on certain curation may enable learning in more generic settings. Evaluating such methods with those under substantially different assumptions risks discouraging precisely the kinds of exploratory research that unsupervised learning has historically enabled.

**Supporting Discovery at Comparable Scale.** Finally, we emphasize the importance of enabling scientific discovery at comparable scales. Large pre-trained models require industrial-level resources on data and compute that are inaccessible to much of the research community. As a result, exploration of alternative learning paradigms becomes more difficult in the same scale. Encouraging research that operates at similar data and computational scales, while varying assumptions and learning signals, allows diverse ideas to be evaluated on more equal footing. This is particularly important when novelty matters more than achieving state-of-the-art performance. By supporting discovery at a comparable scale, the community can support a broader and more inclusive range of unsupervised learning research.

**Summary.** Taken together, these proposals aim to improve clarity without restricting innovation. By distinguishing pre-trained-dependent methods, encouraging cross-regime evaluation, disclosing required priors, valuing assumption relaxation, and supporting same-scale exploration, the field

can preserve the empirical successes of modern visual learning while sustaining deeper scientific progress. In practice, such disclosure could be implemented as a short standardized reporting item, analogous in spirit to existing compute or resource reporting practices. The intent is not to create a new evaluation gate, but to make key assumptions visible with minimal overhead. Even a compact disclosure of pre-training source, data scope, augmentation invariances, and adaptation strategy can clarify whether two methods called "unsupervised" are operating under comparable assumptions.

## 6. Alternative Views

**View (1):** *All learning pipelines inevitably involve human choices, rendering distinctions between supervised and unsupervised learning largely semantic. Data must be collected, sensors must be designed, preprocessing steps must be chosen, and objectives must be specified. From this viewpoint, no learning system is free of human assumptions, and without priors to exploit, learning would be impossible.*

We agree that learning without assumptions is neither feasible nor desirable. Machine learning systems rely on inductive biases that reflect some prior on the data. However, acknowledging the inevitability of assumptions strengthens rather than weakens the case for making them explicit. When assumptions remain implicit, it becomes difficult to analyze why a method works, under what conditions it may fail, and how it compares to alternative approaches. Explicitly stating assumptions does not diminish a method's contribution; rather, it enables clearer scientific communication, which accelerates cumulative progress built on shared grounding. In this sense, our proposal is not to eliminate priors, but to render them visible and discussable.

**View (2):** *As long as the assumptions align with the target data distribution, exploiting such priors should be encouraged. For example, large-scale image data crawled from the internet naturally exhibits object-centric framing, photog-*

*rapher bias, and categories. These are the exact properties that make web-scale data both abundant and practically useful, and leveraging them has led to substantial empirical success.*

We fully agree that exploiting such regularities is often sensible and productive; many recent advances in visual learning depend precisely on aligning learning objectives with these widely available data characteristics. Our argument is not that such priors are illegitimate or should be avoided. Rather, methods that rely on them should be described in ways that accurately reflect their dependencies. When all label-free methods are grouped under a single "unsupervised" umbrella, important differences in data assumptions and learning regimes are obscured. By explicitly stating which priors are being exploited, such as object-centricity, augmentation invariance, or taxonomic alignment, researchers can communicate the scope and limitations of their methods more clearly, without diminishing their practical value.

**View (3):** *Empirical performance, rather than conceptual categorization, should be the primary criterion for evaluating learning methods. From this perspective, if a method performs well across a wide range of benchmarks, the nature of its supervision or assumptions is of secondary importance.*

While empirical performance is undeniably central to progress, we argue that performance alone is insufficient for understanding. Without clarity about underlying assumptions, strong results may not translate across domains, data regimes, or problem settings. Explicit disclosure of assumptions enables more accurate interpretation of empirical results and helps anticipate where methods are likely to generalize or break down. Conceptual clarity and empirical success are not competing goals; rather, clarity provides the context needed to interpret performance meaningfully.

**View (4):** *Increasing focus on assumptions risks fragmenting the field or complicating evaluation standards.*

We believe the opposite is more likely. By clarifying assumptions, researchers can better position their contributions within a broader landscape of learning paradigms, allowing complementary approaches to coexist and be evaluated on their intended terms. Rather than narrowing the field, such clarity supports a richer ecosystem of methods addressing diverse settings, from curated large-scale datasets to low-resource or unconventional visual domains.

**Summary.** Taken together, these alternative perspectives reinforce our central claim: assumptions are unavoidable, useful, and often beneficial. The question is not whether to use priors, but how clearly they are articulated. By encouraging explicit disclosure of data priors and pipeline dependencies, the community can preserve the empirical strengths of modern visual learning while improving conceptual clarity, comparability, and long-term scientific progress.

## 7. Conclusion

The past decade has demonstrated that remarkably powerful visual representations can be learned without human labels. Large-scale self-supervised pre-training has become foundational infrastructure for modern vision systems, and its empirical success is undeniable. However, *the absence of human labels does not mean that human supervision is absent*, and conflating these two notions risks obscuring the scientific foundations of unsupervised learning. Our argument is not that unsupervised methods exploiting human biases or priors are less valid, nor that such priors should be avoided. Rather, our central claim is scientific: human supervision enters through data curation and pre-processing choices, and invariances encoded in human-selected learning objectives. When these assumptions remain implicit, it becomes difficult to disentangle the contribution of a new learning mechanism from inherited priors embedded in the data or pre-trained representations.

Our empirical analysis provides a community-level signal of this shift. Title-based statistics show that explicit use of the terms "unsupervised" and "self-supervised" peaks around 2021 and declines thereafter, despite continued growth in publication volume. A complementary full-text analysis of *CVPR* papers follows the same qualitative pattern, suggesting that this is not merely a title-level naming artifact. At the same time, among *CVPR* papers titled with "unsupervised," reliance on pre-trained representations increases over time. Together, these patterns suggest that unsupervised learning has become simultaneously successful and less explicitly articulated: its techniques are widely adopted, yet the assumptions that enable them are less often foregrounded. In our view, this ambiguity risks misattributing emergent behavior in unsupervised models, distorting evaluation incentives, and narrowing methodological diversity.

To address these risks, we propose lightweight practices aimed at improving clarity without restricting innovation. These include distinguishing pre-training-dependent pipelines from methods that learn from raw data, encouraging robustness across pre-training regimes, explicitly disclosing data and augmentation priors, valuing assumption relaxation as a scientific contribution, and supporting comparison at a comparable computational scale. Together, these practices aim to preserve what has historically made unsupervised learning scientifically productive: making assumptions explicit and testing what can be relaxed.

As a concluding remark, the field does not need less large-scale self-supervised pre-training, but would benefit from clearer scientific language about where supervision resides. By making assumptions and dependencies explicit, the computer vision research community can better trace sources of supervision and sustain more transparent long-term progress in unsupervised learning.

## Acknowledgements

The author thanks Stefano Soatto (UCLA), Alex Wong (Yale), Yanchao Yang (HKU), Bolei Zhou (UCLA), Hao Wang (Rutgers), Francesco Locatello (ISTA), Jiantao Wu (University of Surrey), and Naiyan Wang (Xiaomi) for valuable discussions and feedback. Their comments helped sharpen the framing of this paper, although the manuscript does not necessarily reflect their views or positions. Indeed, these discussions included perspectives that differed from, and sometimes directly conflicted with, one another. This disagreement further reinforces the importance of open discussion around how supervision is defined, disclosed, and evaluated in modern visual learning.

The author thanks Yi Xiao (LSU), Austin Louque (LSU), and Ahmad Hijazi (LSU) for assistance with the manual annotation of pre-training and supervision dependencies. The work is supported by the author's startup funds at LSU.

## Impact Statement

This work discusses how human supervision enters unsupervised deep learning models through data and learning objectives. We do not foresee negative societal impacts. The proposed approaches allow researchers with limited compute, including those from economically constrained regions, better participate in unsupervised learning research.

## LLM Statement

The authors used LLMs for grammar and proofreading. All arguments were developed by the authors, who remain fully responsible for the content.

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
