# OpenReview forum: "Position: Unlabeled ≠ No Human Supervision in Visual Learning"
_ICML.cc/2026/Position_Paper_Track — ICML 2026 Position Paper Track regular_

### Official Review · Reviewer_xDuf · 2026-03-11

**Significance:** 3
**Argument Clarity:** 3
**Rating:** 4
**Confidence:** 3

**Questions:**

1. While using paper title analysis as a proxy is useful, wouldn't it be more accurate to use LLM analysis on the abstract or methods section of a paper and then manually verify the results to determine whether genuine unsupervised learning research is truly decreasing?

2. Based on the opinion of the paper, if the data collection process itself involves human selection, and the objective function also includes human priors, then does the concept of ``unsupervised learning'' completely lose its meaning?

3. If the community is to adopt this checklist, how can it be actually implemented to ensure it truly improves scientific communication and doesn't become mere formality and superficial compliance?

**Alternative Views Section:**

Yes

**Compliance With Llm Reviewing Policy A Conservative:**

Affirmed.

**Discussion Potential:**

3

**Final Justification:**

Thanks for the rebuttal to solve my concern. I think this is a meaningful and valuable paper. I have no further questions at present. I maintain my score.

**Paper Summary:**

This position paper argues that, in visual learning, not using the labels in the training set is not equivalent to not using human supervision.
The authors observe that despite the continued growth in the number of papers published in CV community, the explicit use of the term ``unsupervised'' in the titles of papers at top computer vision conferences (such as CVPR, ICCV, and ECCV) peaked around 2021 and then began to decline.
Meanwhile, methods claiming to be ``unsupervised'' increasingly rely on a centralized set of pre-trained base models (e.g., DINO, CLIP, and MoCo).
This paper argues that human supervision is implicit in data management, prior knowledge of dataset distribution (e.g., object-centric frameworks), and learning objectives. To maintain methodological diversity and conceptual clarity, the authors propose a standardized disclosure checklist, urging researchers to state their data biases and pre-training dependencies explicitly.

**Position:**

Yes

**Position In Title:**

Yes

**Related Work:**

2

**Strengths And Weaknesses:**

**Strengths**:

1. This paper points out a relatively common issue that is often acknowledged but seldom addressed in current representation learning. The author(s) clearly distinguish between unlabeled learning and learning without assumptions or human intervention.

2. This paper does not merely critique mainstream foundational model paradigms, but rather offers a practical and lightweight list of disclosures designed to increase transparency without penalizing the use of strong priors or stifling innovation.

**Weaknesses**:
1. This paper's empirical analysis relies heavily on keyword statistics from the paper's title. However, such indicators may only reflect some ``marketing trends'' or changes in naming conventions within academia, and do not necessarily represent a significant shift in scientific methodology.

2. The paper offers several explanations for the decrease in the use of the term "unsupervised": Supply-side pressure, Demand-side reframing, and User-side indifference. These explanations are logically plausible, but they are essentially speculations. But lack empirical evidence to prove that these reasons actually cause this phenomenon.

**Support:**

3

---

> ### Author Rebuttal · Authors · 2026-03-28
>
> We thank the reviewer for the thoughtful and constructive feedback, and for recognizing both the importance of the distinction and the practical value of the proposed checklist.
>
> 1. *“Title-based analysis may reflect naming trends rather than methodology”*
>
> As suggested by the reviewer (also by Reviewer iMd1), we extended the analysis to the full-text level across all 17,435 CVPR papers (2015–2025):
>
> | Year | # Papers (Un/Self-Sup) | Total Papers | % Papers |
> |------|------------------------|--------------|----------|
> | 2015 | 61  | 602  | 10.13% |
> | 2016 | 56  | 643  | 8.71%  |
> | 2017 | 82  | 783  | 10.47% |
> | 2018 | 126 | 979  | 12.87% |
> | 2019 | 190 | 1294 | 14.68% |
> | 2020 | 255 | 1466 | 17.39% |
> | 2021 | 368 | 1660 | **22.17%** |
> | 2022 | 433 | 2071 | 20.91% |
> | 2023 | 509 | 2353 | 21.63% |
> | 2024 | 421 | 2713 | 15.52% |
> | 2025 | 369 | 2871 | 12.85% |
>
> This matches the pattern described in the paper, and we will incorporate these results in the revised version. The rationale behind the title-based analysis is to deliberately keep the signal coarse, without engaging in potentially contentious distinctions such as “what counts as unsupervised/self-supervised” or how individual authors define their methods. In addition, titles reflect how authors choose to foreground the unsupervised aspect of their work, making them a meaningful indicator of community-level positioning.
>
> In the revised paper, we will present both title-based analysis and full-text analysis, providing a complementary view that captures both how methods are framed and how they are implemented.
>
> For the accuracy of the LLM-based paper analysis, we refer to our response to Reviewer iMd1, where we conducted manual inspection and annotation of all 287 CVPR papers (2020–2025) with “unsupervised” in the title, demonstrating strong reliability of the analysis.
>
> 2. *“Hypotheses are speculative”*
>
> The hypotheses in the paper are intended to be descriptive rather than causal, and we will clarify this more explicitly in the revision. In the manuscript, these hypotheses are presented as possible interpretations of the observed trend, not as definitive explanations.
> This is consistent with the broader framing of the paper: we first establish that a systematic shift has occurred in how unsupervised learning is positioned and practiced in computer vision, and only then discuss several plausible factors that may contribute to this shift. A more conclusive evaluation of these hypotheses would require a community-level study, such as detailed case analyses or large-scale surveys of authors, which is beyond the scope of the current manuscript.
>
> We will revise the paper to make this distinction clearer and avoid any potential over-interpretation.
>
> 3. *“Does unsupervised learning lose its meaning?”*
>
> Please refer to Section 6 (Alternative Views, Point 1) for a detailed discussion. Our position is that unsupervised learning is at risk of becoming under-specified in modern pipelines, a shift that is also reflected in our community-level analysis.
>
> As discussed in the paper, earlier unsupervised learning operated in-domain, where assumptions were explicit and localized. In contrast, modern pipelines rely on foundation-scale pretraining and transfer, where supervision enters but is not articulated.
>
> The issue is therefore not that unsupervised learning is invalid, but that methods labeled as “unsupervised” can differ substantially in their sources of supervision, while these differences are not systematically communicated. As noted in the paper, this ambiguity can lead to misattribution of emergent behavior, erosion of conceptual clarity, and distorted evaluation of methods.
>
> Our goal is thus not to redefine unsupervised learning, but to restore its interpretability by making underlying assumptions explicit and visible.
>
> 4. *“How can it be actually implemented?”*
>
> The checklist is designed to be minimal yet structured, so that it can be standardized and potentially adopted by conferences without imposing significant overhead on authors. A similar precedent exists in "Compute Reporting Form" at CVPR, where authors disclose computational scale and hardware to improve transparency.
> As demonstrated in our response to Reviewer iMd1, we provide instantiations showing that the checklist captures meaningful differences in practice while remaining simple to apply. The goal is not to enforce a rigid framework, but to introduce a lightweight mechanism for making key assumptions visible.
>
> More broadly, we view this checklist as an initial step rather than a final solution. As the field continues to evolve, more refined or standardized approaches may emerge. However, we believe it is important to take the first step.
>
>
>
> We hope these additions address the reviewer’s concerns, and they will be reflected in the updates of the main paper. We are happy to further clarify any aspect and hope the reviewer will take these additional analyses into account in their final assessment.

---

> > ### Author Rebuttal · Reviewer_xDuf · 2026-04-01
> >
> > Overall, I think this is a meaningful and valuable paper, and I have no further concerns at this point.

---

### Official Review · Reviewer_TxEL · 2026-03-12

**Significance:** 2
**Argument Clarity:** 3
**Rating:** 4
**Confidence:** 4

**Questions:**

See above

**Alternative Views Section:**

Yes

**Compliance With Llm Reviewing Policy A Conservative:**

Affirmed.

**Discussion Potential:**

3

**Final Justification:**

------------------After rebuttal ---------------
The authors have partially resovled my concerns. My opinion is acutually currectly borderline, but slightly leans positively. So i give a borderline accept.

**Paper Summary:**

This position paper argues that "unsupervised" visual learning is not truly "human-free" and urges the research community to explicitly disclose the human priors embedded in data curation, pre-processing, and learning objectives. Through an analysis of flagship computer vision conferences (CVPR, ICCV, ECCV), the authors demonstrate that while the explicit use of terms like "unsupervised" and "self-supervised" in paper titles has declined since 2021, the reliance on a small set of dominant pre-trained backbones (e.g., DINO, CLIP) has significantly increased. To improve conceptual clarity and preserve methodological diversity, the authors propose a standardized disclosure checklist for label-free visual learning claims

**Position:**

Yes

**Position In Title:**

Yes

**Related Work:**

2

**Strengths And Weaknesses:**

Strengths:
1. Timely and Relevant Critique: The paper addresses a critical shift in the field where "unsupervised" methods increasingly depend on foundation models, potentially obscuring the true source of performance gains.
2. Empirical Grounding: The authors provide quantitative evidence of community-level trends using title-based statistics and an LLM-based analysis of manuscript dependencies from 2013 to 2025.
3. Constructive Proposal: Rather than merely critiquing current practices, the paper offers a concrete "Disclosure Checklist" (Table 1) to help researchers distinguish between label-free learning and the absence of human supervision.
4. Nuanced Perspective: The authors acknowledge that human priors are unavoidable and often beneficial; their goal is transparency rather than the elimination of these priors.

Weaknesses:
1. The major weakness in my view is that the major texts stated by the paper seems to be little related with the title. In the title, it states that Unlabeled /= No Human Supervision in Visual Learning. However, in the proposal of this paper, the authors seems to mostly overlook the contents of the title, i.e., Human Supervision, and try to analyze how unsupervised and self-supervised learning should develop. Actually, i don't grap strong correlations between No Human Supervision and the proposal which describes the way unsupervised and self-supervised learning develop. This seems to be seperated from the proposal where most of the proposed ideas including Valuing Assumption Relaxation, Supporting Discoveryat Comparable Scale and Explicit Disclosure of Required Priors could be directly presented without previous heavy foreshadow.

2. Besides, another weakness in my view is that while the authors do introduce a new concept that Unlabeled /= No Human Supervision, knowing Human Supervision having participating in the data curation process doesn't strongly change the current research field, or how unsupervised/self-supervised methods are developped. It's also weakly related with the proposal in the paper, which can be presented without the preceding analysis.

3. While the paper argues that data curation is a form of supervision, the boundary between "inherent natural priors" and "human-introduced biases" can be difficult to define in practice.

4. The analysis of pre-training dependence (Figure 3) is limited to a relatively small set of well-known and ole families, potentially overlooking emerging or niche pre-training regimes. In this paper, few state-of-the-art architecture or methods are discussed.

**Support:**

3

---

> ### Author Rebuttal · Authors · 2026-03-28
>
> We thank the reviewer for the thoughtful reading and for recognizing the timeliness, empirical grounding, and constructive nature of the paper.
>
> 1. *Connection between title and proposal*
>
> We thank the reviewer for raising this point and clarify the logical connection more explicitly.
>
> The title states the central argument, grounded in our empirical community-level observations: **“unlabeled” does not imply the absence of human supervision**. In modern pipelines, supervision enters through multiple channels, even when labels are absent. As shown in the paper, there is a clear community-level shift in how “unsupervised” learning is positioned, reflected by both the declining prominence of the term and the increasing dependence on pre-training among "unsupervised" methods (see response to Reviewer iMd1).
>
> The core issue we highlight is that such methods are grouped under the same umbrella term “unsupervised,” without distinguishing their underlying sources of supervision. This lack of differentiation can obscure what drives performance and places unsupervised research at risk of losing conceptual clarity. We thus adopt a direct title to emphasize this point. If helpful, we are open to refining the title (e.g., with a subtitle such as “the community needs more transparency on the source of supervision”) to further clarify this connection.
>
> The proposal follows directly from this observation. Once supervision is no longer localized in labels but instead distributed across the pipeline, it becomes necessary to make these sources of supervision explicit in order to understand what drives performance and to enable fair comparison. In this sense, the proposal is not separate from the title claim, but a direct consequence of it. We will revise the paper to make this connection clearer.
>
> 2. *“Does not strongly change the field / weak impact”*
>
> We clarify that the goal of the paper is not to directly change how methods are developed, but to advocate for transparency in how they are understood and evaluated. Transparency does not immediately alter methodology, but it provides the foundation for clearer attribution of performance gains, more meaningful comparisons, and ultimately more informed progress.
>
> At the same time, the issue we identify is not purely conceptual. As shown in our updated analysis (see responses to R1 and R4), unsupervised learning is at risk at the community level, with both its relative proportion and absolute count declining significantly since its peak around 2021 at CVPR. This suggests that the way unsupervised learning is framed and practiced is undergoing a structural shift. To our knowledge, this is the first work documenting this trend at scale, which we believe is itself a meaningful contribution.
>
> We also acknowledge that it is not realistic for a single position paper to directly change the trajectory of the field. Our intention is to take an initial step to surface this issue and spark broader discussion. As the reviewer noted, the discussion potential of this work is good, and we view this as an encouraging signal.
>
> 3. *Boundary between natural priors and human bias*
>
> We agree that the boundary between “natural priors” and “human-introduced biases” can be difficult to define in practice. However, this difficulty is precisely why explicit disclosure is needed. Rather than attempting to enforce a strict or universal definition, the proposed checklist focuses on making intervention points visible, such as dataset curation, viewpoint selection, and augmentation assumptions. The goal is not to perfectly categorize all sources of prior knowledge, but to make them explicit and discussable, so that their role can be better understood in context.
>
> 4. *Limited coverage of pretraining methods*
>
> Since the study aggregates papers from 2020 to 2025, more recent methods naturally appear less frequently due to limited time for adoption. We clarify that the observed concentration reflects the pattern in the data, rather than a limitation of the analysis itself.
>
> More importantly, we deliberately refrain from analyzing individual methods in detail. As stated in the paper, the goal is not to evaluate or “call out” specific works, but to identify community-level trends. Going deeper into individual methods, especially emerging or niche pre-training regimes, is not the focus of this paper and can easily lead to case-by-case debates or controversy. Instead, we aim to establish a common ground for discussion by focusing on aggregate patterns.
>
> We hope these clarifications address the reviewer’s concerns and better establish the connection between the title, analysis, and proposal. We are happy to further clarify any aspect and hope the reviewer will take these additional discussions, as well as the updated empirical analysis, into account in their final assessment. We will incorporate the discussions into the main paper per requested by the reviewer.

---

> > ### Author Rebuttal · Reviewer_TxEL · 2026-04-02
> >
> > The rebuttal has resovled most of my concerns, but i still have several remaining points. Regarding the trend of why unsupervised and self-supervised start to decline since 2021, i think that in the perspective of engineering, adopting unsupervised and self-supervised learning for smaller models is easier, while current in the age of LLMs and MLLMs, adopting unsupervised and self-supervised learning for largers models is more difficult and hard to produce performance gains than increasing data or parameter size. This may not strictly correspond the shift of unsupervised and self-supervised learning, but relates with acturally performance considerations.

---

### Official Review · Reviewer_izdp · 2026-03-12

**Significance:** 2
**Argument Clarity:** 2
**Rating:** 4
**Confidence:** 4

**Questions:**

Please refer to the weaknesses.

**Alternative Views Section:**

Yes

**Compliance With Llm Reviewing Policy A Conservative:**

Affirmed.

**Discussion Potential:**

2

**Final Justification:**

I do not believe my concerns were fully addressed in the rebuttal. The information provided by the authors did not go beyond the scope of the original paper, nor did it alter my perception of the work. However, I acknowledge that the topic raised is of significant interest, particularly in the era of Large Language Models (LLMs). The priors and knowledge embedded in LLMs have transcended the initial definition of supervision, and our community indeed needs to pay attention to this issue. Given this, I am willing to raise my score to encourage more attention to this topic.

**Paper Summary:**

This paper argues that the absence of labels in visual learning does not imply the absence of human supervision. It therefore calls on the research community not to equate unlabeled methods with unsupervised learning. The authors first analyze the usage trends of the terms “unsupervised” and “self-supervised” in paper titles from top computer vision conferences (CVPR, ICCV, ECCV). They find that the usage of these terms peaked around 2021 and then unexpectedly declined. This suggests that the computer vision community has undergone a shift in how it positions and discusses unsupervised learning.
The paper then explains the risks of conflating unlabeled learning with learning without human supervision, including Misattribution of Emergent Behavior, Erosion of Conceptual Clarity, Distorted Research Incentives, Reduced Technical and Methodological Diversity, and Barriers to Same-Scale Evaluation. The authors further analyze how human supervision can implicitly influence unlabeled data. Finally, they propose several recommendations, including distinguishing methods that rely on pretraining, explicitly disclosing required priors, conducting robustness tests across different pretraining mechanisms, emphasizing the relaxation of assumptions, and supporting discovery at comparable scales.

**Position:**

Yes

**Position In Title:**

Yes

**Related Work:**

2

**Strengths And Weaknesses:**

Strengths

The paper is well structured and clearly written.

The paper strengthens its argument by conducting an empirical analysis of terminology trends in paper titles from top computer vision conferences, rather than relying solely on theoretical discussion. This quantitative evidence makes the claims more convincing.

The paper advocates for clearer disclosure of assumptions, greater methodological diversity, and research at comparable scales in order to foster a healthier research ecosystem and encourage fundamental innovation.

Weaknesses
The paper’s central claim that the absence of labels in visual learning does not imply the absence of human supervision does not appear to be a viewpoint that necessarily requires extensive argumentation. Similar ideas have been discussed in philosophy. For example, Immanuel Kant argued that our perception of the world is shaped by human cognitive structures: “The world we perceive is processed through the structures of human cognition.” In this sense, whether learning is unsupervised or self-supervised, it is still ultimately grounded in human cognition.

Although I agree with the authors’ attempt to clarify and broaden the notion of unsupervised learning, the proposed suggestions do not constitute a systematic method or framework that can effectively guide subsequent research. For instance, many studies already compare methods using different pretrained models and models with varying parameter scales during benchmarking and ablation studies. This observation somewhat weakens the practical contribution of the paper.

In addition, the recommendation “Explicit Disclosure of Required Priors” is rather coarse-grained. The paper should more systematically specify what types of priors need to be disclosed, so that future researchers can more easily follow and implement the proposed guidelines.

**Support:**

2

---

> ### Author Rebuttal · Authors · 2026-03-28
>
> We thank the reviewer for the detailed feedback and for recognizing the clarity of writing and the empirical grounding of the paper.
>
> 1. *“The central claim is philosophically known and does not require extensive argumentation”*
>
> We respectfully disagree with this assessment.
>
> While the general idea that human cognition shapes perception has long been discussed in philosophy, the contribution of this paper is not philosophical, but **field-specific and empirical**. As also noted by other reviewers, this issue is timely (TxEL), commonly observed yet seldom explicitly addressed (xDuf), and “spot on” (iMd1).
>
> What is new here is not the abstract statement that “priors exist,” but the systematic documentation of how modern visual learning pipelines operationalize these priors, and how this differs from earlier formulations of unsupervised learning. In particular, our analysis shows a clear community-level shift: while “unsupervised” methods are less explicitly foregrounded, their dependence on pretraining is rapidly increasing. This is not a philosophical argument, but an **empirical phenomenon**, which, to our knowledge, has not been previously quantified at this scale.
>
> The paper therefore connects a conceptual observation with a measurable community-level trend and a practical implication for how results should be interpreted. It is precisely this connection that motivates the proposal.
>
> 2. *“The proposal does not form a systematic framework or strongly guide research”*
>
> We clarify that the goal of the paper is not to prescribe how research should be conducted, nor to introduce a rigid framework that constrains methodology. Instead, the goal is to promote **clarity** in how methods are understood and evaluated.
>
> The reviewer’s observation that many works compare different pretrained models or parameter scales is valid, and we agree that such comprehensive studies are increasingly common. In fact, we view this as supporting our central argument rather than contradicting it. These studies demonstrate that different pre-training choices can lead to substantially different behaviors and performance, even when methods are all described as “unsupervised” or “self-supervised.” This observation makes it even more important to differentiate and clarify the source of supervision.
>
> However, the differences documented by such studies are not always systematically reflected in how methods are communicated. As a result, methods with distinct sources of supervision may still be grouped under the same umbrella term, which can make interpretation and comparison less clear.
>
> The proposal therefore aims to make these factors more explicit and consistently reported, so that such differences, which are already observed in practice, can be more transparently accounted for. While transparency does not directly dictate how research is conducted, it provides a clearer foundation for comparison and interpretation, and can ultimately contribute to more informed and meaningful progress.
>
> 3. *“The disclosure of priors is too coarse-grained”*
>
> We agree that this is an important consideration.
>
> The checklist is intentionally designed to be **minimal yet structured**, so that it can be broadly adopted without imposing significant overhead. There is a natural tension between being overly coarse and overly rigid (also discussed in our response to Reviewer xDuf). Our design aims to strike a balance. We appreciate the reviewer’s perspective on this point, and if there are specific suggestions for refining the level of detail or structure, we would be happy to incorporate them in the revision. At the same time, we view the checklist as an initial step toward improving transparency, and we believe that potential refinements do not affect the central empirical observations or the overall motivation of the paper.
>
> At the same time, we provide concrete instantiations (in response to Reviewer iMd1), demonstrating that the current checklist can capture meaningful distinctions in practice. Again, we view this proposal as an initial step rather than a final taxonomy. More refined or standardized formulations may emerge as the community adopts and builds upon this direction.
>
> *Remarks*
>
> We believe there may be some difference in interpretation regarding the intended scope of the paper. Our primary goal is not to propose a new methodological framework, but to present a field-grounded empirical observation and take a practical step toward improving transparency in how methods are understood and compared.
>
> With the addition of full-text validation and manual verification (see responses to Reviewers iMd1 and xDuf), the empirical foundation of the paper has been further strengthened. We hope these clarifications help better convey the intent and contribution of the work, and we sincerely appreciate the reviewer’s thoughtful feedback. We are happy to further discuss any aspect and hope these additions will be taken into consideration in the final assessment.

---

> > ### Author Rebuttal · Reviewer_izdp · 2026-04-03
> >
> > I do not believe my concerns were fully addressed in the rebuttal. The information provided by the authors did not go beyond the scope of the original paper, nor did it alter my perception of the work. However, I acknowledge that the topic raised is of significant interest, particularly in the era of Large Language Models (LLMs). The priors and knowledge embedded in LLMs have transcended the initial definition of supervision, and our community indeed needs to pay attention to this issue. Given this, I am willing to raise my score to encourage more attention to this topic.

---

### Official Review · Reviewer_iMd1 · 2026-03-15

**Significance:** 3
**Argument Clarity:** 3
**Rating:** 5
**Confidence:** 4

**Questions:**

See Weaknesses section.

**Alternative Views Section:**

Yes

**Compliance With Llm Reviewing Policy A Conservative:**

Affirmed.

**Discussion Potential:**

4

**Final Justification:**

Thanks for the thorough response. The rebuttal addresses my main concerns. The checklist examples for DINO/MAE are helpful too. Raising my score to Accept.

**Paper Summary:**

The paper argues that "unsupervised" in visual learning is misleading because it conflates "no labels" with "no human supervision." In reality, data curation, filtering, object-centric framing, augmentation choices etc. all inject human priors into supposedly unsupervised pipelines. The authors do a title-based analysis of CVPR/ICCV/ECCV (2013-2025) showing the term "unsupervised" peaked around 2021 then declined, while pre-training dependence keeps going up. They also show that a small number of backbones dominate. They propose a disclosure checklist (Table 1) to make these dependencies explicit.

**Position:**

Yes

**Position In Title:**

Yes

**Related Work:**

3

**Strengths And Weaknesses:**

Strengths
1. I think the core observation is spot on. The gap between what "unsupervised" implies and what actually happens in modern pipelines is real, and this paper articulates it well. The framing is clean: label-free != supervision-free.
2. More empirical effort than most position papers. The title trend analysis (Figure 1), pre-training dependence trend (Figure 2), and backbone concentration give the argument concrete grounding rather than just vibes.
3. Alternative views section (Section 6) is well done. The counterarguments are real, and the responses are measured rather than dismissive.

Weaknesses
1. The title keyword analysis is the main quantitative evidence but it's pretty shaky. Titles reflect naming conventions, not methodology. The decline of "unsupervised" in titles could just mean people switched to saying "foundation model" or "self-supervised pre-training." A paper called "Masked Image Modeling for Visual Representation Learning" is self-supervised but wouldn't show up. I think some validation against abstracts or method sections is needed to know whether this is a real signal or just a fashion trend.
2. The LLM-based classification or Figure 2 has no reported validation. No precision/recall, no comparison with human labels. I'm not sure a 7B model can reliably tell the difference between "uses a frozen CLIP backbone" vs "compares against CLIP as a baseline," which are very different things.
3. I wish the authors had filled out their own checklist (Table 1) for at least one known method like DINO or MAE. Would make the proposal much more concrete and also reveal practical difficulties.
4. Minor: the scope is vision only, which is fine, but the argument likely applies to other modalities too. Even briefly mentioning that would make the position feel more general.

**Support:**

3

---

> ### Author Rebuttal · Authors · 2026-03-28
>
> We thank the reviewer for the support, and for recognizing both the central distinction and the effort to ground this position paper with empirical analysis.
>
> 1. *“Titles reflect naming conventions, not methodology…”*
>
> In the paper, we explicitly adopt the title-based statistics as a **conservative** signal of how the field positions and foregrounds unsupervised learning. We deliberately avoid paper-level judgments or “calling out” individual works (as noted in the footnote of the paper), and instead focus on community-level patterns, which we believe is appropriate for a position paper.
>
> To directly address this concern, we followed the reviewer’s suggestion and extended the analysis to the full-text level across all 17,435 CVPR papers (2015-2025). The same rise-and-decline pattern persists:
>
> | Year | # Papers (Un/Self-Sup) | Total Papers | % Papers |
> |------|------------------------|--------------|----------|
> | 2015 | 61  | 602  | 10.13% |
> | 2016 | 56  | 643  | 8.71%  |
> | 2017 | 82  | 783  | 10.47% |
> | 2018 | 126 | 979  | 12.87% |
> | 2019 | 190 | 1294 | 14.68% |
> | 2020 | 255 | 1466 | 17.39% |
> | 2021 | 368 | 1660 | **22.17%** |
> | 2022 | 433 | 2071 | 20.91% |
> | 2023 | 509 | 2353 | 21.63% |
> | 2024 | 421 | 2713 | 15.52% |
> | 2025 | 369 | 2871 | 12.85% |
>
> This confirms what we observed in the paper: the unsupervised learning is increasingly less explicitly foregrounded since its peak around 2021, despite the continued growth of the field. This confirms that the trend is not a naming artifact, but a consistent community-level shift.
>
> 2. *Validation of LLM-based classification*
>
> To address the concern regarding the reliability of the LLM-based analysis, we performed **manual** inspection of all 287 CVPR papers (2020-2026) with "unsupervised" in the title. Below is the accuracy of LLM-based analysis:
>
> | Metric     | Value  |
> |------------|--------|
> | Precision  | 91.5%  |
> | Recall     | 81.8%  |
> | Accuracy   | 86.9%  |
>
> The model achieves high precision, meaning that when it identifies pre-training dependence, it has a high accuracy. The lower recall reflects a conservative bias, primarily missing weaker or indirect dependencies (mostly implicit ImageNet initialization). As a result, the measured trend is more likely a lower bound. Upon manual annotation, the increase in pre-training dependence among unpervised papers remains extremely clear:
>
> | Year | Pretraining Dependency |
> |------|------------------------|
> | 2020 | 32.5% |
> | 2021 | 39.0% |
> | 2022 | 45.2% |
> | 2023 | 53.5% |
> | 2024 | 62.5% |
> | 2025 | **72.4%** |
>
> 3. *Instantiating the checklist (DINO / MAE)*
>
> Below we instantiate the checklist for representative pipelines:
>
> **Example: a hypothetical pipeline using DINO pre-training for semantic segmentation**
>
> | Dimension                  | Description |
> |---------------------------|------------|
> | Pre-training dependence   | Yes (DINO, ViT-B backbone, 4 consumer-grade GPUs) |
> | Frozen vs adaptive        | frozen (zero-shot); fully fine-tuned (best practice) |
> | Pre-training data scope   | ImageNet (~1.4M images) |
> | Data distribution priors  | Object-centric images; 1k predefined semantic categories; balanced class distribution |
> | Learning invariances      | Strong augmentation invariance (multi-crop, photometric transforms, occlusion) |
> | Leakage risks             | Potential overlap between pretraining categories and downstream object of interest |
>
> **Example: a hypothetical action recognition pipeline using MAE pre-training on YouTube videos**
>
> | Dimension                  | Description |
> |---------------------------|------------|
> | Pre-training dependence   | Yes (MAE, ViT backbone, 32 H200 GPU cluster) |
> | Frozen vs adaptive        | Fine-tuned |
> | Pre-training data scope   | 10M YouTube video clips, 20 seconds each |
> | Data distribution priors  | User-uploaded videos, described as human actions|
> | Learning invariances      | Reconstruction via masking; implicit spatial/semantic assumptions |
> | Leakage risks             | No data overlap (verified), overlapping action categories |
>
> These examples reflect a minimum intervention in the current research pipeline, yet they illustrate the core point: both are labeled “unsupervised,” yet they rely on strong and distinct priors, which are currently not systematically disclosed.
>
>
> 4. *Scope*
>
> We agree with the reviewer that the empirical analysis focuses on computer vision. To our knowledge, our paper is the first systematic documentation of pre-training dependence at the community level. As computer vision has one of the largest and most active communities where these trends are clearly observable, we view this as a starting point and hope it can spark similar analysis and discussion in other domains.
>
> We hope these additions address the reviewer’s concerns, and they will be reflected in the updates of the main paper. We are happy to further clarify any aspect and hope the reviewer will take these additional analyses into account in their final assessment.

---

> > ### Author Rebuttal · Reviewer_iMd1 · 2026-04-03
> >
> > The full-text analysis and LLM validation numbers address my main concerns. The checklist examples for DINO/MAE are helpful too.

---

### Decision · Program_Chairs · 2026-04-30

**Decision:**

Accept (regular)

**Comment:**

Based on reviews and rebuttal, I'm proposing acceptance.

The paper clearly checks all the aspects we would expect for a position paper: 1) it raises an important point with based on sufficient evidence; 2) it proposes a fairly actionable approach for the community to resolve the "unsupervised/supervised" issues; 3) it properly discussed alternative views.

Re 1. Some reviewers raised concerns on the significance of the publication trends analysis. While how much the analysis is a clear indicator to support the position in the paper remain somehow debatable (see comments from Reviewer xDuf and TxEl), I believe the refined results in the rebuttal do provide enough grounding. Reviewer izdp also pointed out that the overall position is somehow known and it may not need much argumentation. While I may agree at a certain level of abstraction, I think it is important to instantiate the general position in the current practices of ML, which the paper elaborates on correctly.

Re 2. Some reviewers are not fully convinced of the level of detail of the checklist proposed by the authors (see Reviewer izdp). While it can certainly refined, I agree with the authors that there is a delicate trade-off between a coarse framework that is simple enough to be adopted and a finer one, that may be too complex for researchers to engage with it.